# Immunotherapy in Hepatocellular Cancer Patients with Mild to Severe Liver Dysfunction: Adjunctive Role of the ALBI Grade

**DOI:** 10.3390/cancers12071862

**Published:** 2020-07-10

**Authors:** David J. Pinato, Takahiro Kaneko, Anwaar Saeed, Tiziana Pressiani, Ahmed Kaseb, Yinghong Wang, David Szafron, Tomi Jun, Sirish Dharmapuri, Abdul Rafeh Naqash, Mahvish Muzaffar, Musharraf Navaid, Chieh-Ju Lee, Anushi Bulumulle, Bo Yu, Sonal Paul, Neil Nimkar, Dominik Bettinger, Hannah Hildebrand, Yehia I. Abugabal, Celina Ang, Thomas U. Marron, Uqba Khan, Nicola Personeni, Lorenza Rimassa, Yi-Hsiang Huang

**Affiliations:** 1Department of Surgery & Cancer, Imperial College London, Hammersmith Hospital, Du Cane Road, London W120HS, UK; 160311ms@tmd.ac.jp; 2Faculty of Medicine, Tokyo Medical and Dental University, Tokyo 113-8510, Japan; 3Department of Medicine, Division of Medical Oncology, Kansas University Cancer Center, Westwood, KS 66160, USA; asaeed@kumc.edu (A.S.); hhildebrand@kumc.edu (H.H.); 4Medical Oncology and Hematology Unit, Humanitas Cancer Center, Humanitas Clinical and Research Center-IRCCS, 20089 Rozzano, Milan, Italy; tiziana.pressiani@cancercenter.humanitas.it (T.P.); nicola.personeni@hunimed.eu (N.P.); lorenza.rimassa@hunimed.eu (L.R.); 5Department of Gastrointestinal Medical Oncology, The University of Texas MD Anderson Cancer Center, Houston, TX 77030, USA; akaseb@mdanderson.org (A.K.); YIMohamed@mdanderson.org (Y.I.A.); 6Department of Gastroenterology, Hepatology & Nutrition, The University of Texas MD Anderson Cancer Center, Houston, TX 77030, USA; ywang59@mdanderson.org; 7Department of Internal Medicine, Baylor College of Medicine, Houston, TX 77030, USA; DSzafron@mdanderson.org; 8Department of Medicine, Division of Hematology/Oncology, Tisch Cancer Institute, Mount Sinai Hospital, New York, NY 10029, USA; Tomi.Jun@mountsinai.org (T.J.); sirish.dharmapuri@mountsinai.org (S.D.); celina.ang@mssm.edu (C.A.); thomas.marron@mountsinai.org (T.U.M.); 9Division of Hematology/Oncology, East Carolina University, 600 Moye Boulevard, Greenville, NC 27834, USA; abdulrafeh.naqash@nih.gov (A.R.N.); muzaffarm@ecu.edu (M.M.); navaidm@ecu.edu (M.N.); BULUMULLEA17@ecu.edu (A.B.); 10Department of Medicine, Division of Gastroenterology and Hepatology, Taipei Veterans General Hospital and Institute of Clinical Medicine, National Yang-Ming University, Taipei 11217, Taiwan; ssbugi@gmail.com (C.-J.L.); yhhuang@vghtpe.gov.tw (Y.-H.H.); 11Division of Hematology and Oncology, Weill Cornell Medicine/New York Presbyterian Hospital, 1305 York Avenue, Room Y1247, New York, NY 10021, USA; boyu051988@gmail.com (B.Y.); uqk9001@med.cornell.edu (U.K.); 12New York-Presbyterian Brooklyn Methodist Hospital-Weill Cornell Medicine, Brooklyn, NY 11215, USA; sop9033@nyp.org (S.P.); nsn9004@nyp.org (N.N.); 13Department of Medicine II, Faculty of Medicine, Medical Center University of Freiburg, University of Freiburg, 79106 Freiburg, Germany; dominik.bettinger@uniklinik-freiburg.de; 14Department of Biomedical Sciences, Humanitas University, Via Rita Levi Montalcini 4, 20090 Pieve Emanuele, Milan, Italy

**Keywords:** hepatocellular carcinoma, bilirubin, biomarkers, survival, immunotherapy

## Abstract

Immune checkpoint inhibitors (ICI) have shown positive results in patients with hepatocellular carcinoma (HCC). As liver function contributes to prognosis, its precise assessment is necessary for the safe prescribing and clinical development of ICI in HCC. We tested the accuracy of the albumin-bilirubin (ALBI) grade as an alternative prognostic biomarker to the Child-Turcotte-Pugh (CTP). In a prospectively maintained multi-centre dataset of HCC patients, we assessed safety and efficacy of ICI across varying levels of liver dysfunction described by CTP (A to C) and ALBI grade and evaluated uni- and multi-variable predictors of overall (OS) and post-immunotherapy survival (PIOS). We studied 341 patients treated with programmed-death pathway inhibitors (*n* = 290, 85%). Pre-treatment ALBI independently predicted for OS, with median OS of 22.5, 9.6, and 4.6 months across grades (*p* < 0.001). ALBI was superior to CTP in predicting 90-days mortality with area under the curve values of 0.65 (95% CI 0.57–0.74) versus 0.63 (95% CI 0.54–0.72). ALBI grade at ICI cessation independently predicted for PIOS (*p* < 0.001). Following adjustment for ICI regimen, neither ALBI nor CTP predicted for overall response rates or treatment-emerging adverse events (*p* > 0.05). ALBI grade identifies a subset of patients with prolonged survival prior to and after ICI therapy, lending itself as an optimal stratifying biomarker to optimise sequencing of systemic therapies in advanced HCC.

## 1. Introduction

Hepatocellular carcinoma (HCC) carries global mortality of over 600,000 cases every year [1]. The comparatively poorer prognosis of HCC with respect to other malignancies derives from the high proportion of patients presenting with advanced disease, the high rates of recurrence following radical treatment of early-stage tumours and the concomitant presence of liver dysfunction, a factor that often limits aggressive treatment [2,3].

Sorafenib has for a decade remained the only systemic treatment to offer a survival benefit in advanced HCC [4,5]. However, the positive effect of sorafenib is modest, increasing overall survival (OS) from 7.9 to 10.7 months [6], with uncommon objective responses [5], and progression after a median interval of 5.5 months [6]. On the other hand, it is worth noting that multiple levels of evidence underlined how sorafenib therapy might be tailored on patients in order to maximize the therapeutic effects reducing the possibility of adverse events [7], without forgetting the prognostic significance of the adverse events [8]. Although a number of other molecularly targeted therapies have more recently become available in the management of HCC [9], therapeutic resistance limits long-term survivorship and variable patients’ tolerance to kinase inhibitors influences quality of life and access to second-line therapies [10].

Consequently, systemic therapies with non-cross resistant mechanisms of action and non-overlapping toxicity with targeted therapies are highly desirable and urgently needed for the treatment of advanced HCC.

Immune checkpoint inhibitors (ICI) targeting the programmed-cell death-1 receptor/ligand (PD-1/PD-L1) and cytotoxic T-lymphocyte antigen 4 (CTLA-4) pathways [11] have shown initial evidence of anti-tumour activity in HCC. As a consequence, nivolumab and pembrolizumab have been granted conditional approval by the Food and Drug Administration on the basis of non-randomised clinical studies in patients who are intolerant or have progressed to sorafenib [12,13].

The success and clinical positioning of ICI as monotherapies or combinations in HCC are strongly dependent on their ability to improve patients’ survival [14]. Unfortunately, whilst capable of inducing radiologic regression in a little less than 20% of patients, both pembrolizumab and nivolumab, two forerunner PD-1 inhibitors, recently failed to improve overall survival (OS) in advanced HCC, both in first and second line [15]. Combination immunotherapy with atezolizumab and bevacizumab has demonstrated superiority over sorafenib in the first-line treatment of advanced HCC, a finding that is likely to have transformative implications in treatment paradigms of this disease [16].

Unlike most oncological indications, OS is profoundly influenced by liver dysfunction in HCC patients [17]. Traditionally estimated using the Child-Turcotte-Pugh (CTP) classification, liver functional reserve still relies on a numerical score originally developed over 5 decades ago to estimate the peri-operative mortality of cirrhotic patients [18].

Despite being widely used to predict the prognosis of patients with cirrhosis and aid treatment decisions in clinical practice, the CTP score is composed of variables with uneven prognostic ability and variable reliability and reproducibility, especially in patients with superimposed HCC [19].

Recently, the albumin-bilirubin (ALBI) grade, an alternative method of measuring liver function that relies solely on albumin and bilirubin, has been proposed as a biomarker with better clinical utility and reproducibility than the CTP classification [20]. Appealing qualities of the ALBI include the prognostic independence from subjective variables such as clinical grading of ascites and encephalopathy, the capacity to sub-stratify patients within each CTP class according to different survival outcomes and the wide reproducibility across the various stages of HCC [21,22].

Despite the rapidly growing evidence in support of its superior prognostic utility over CTP, the ALBI grade has not been validated in advanced HCC patients receiving immunotherapy [23], a population where more precise liver functional estimate might aid clinicians in identifying the best candidates for this novel therapeutic approach, a point of greater consequence given the negative results from phase III trials in unselected CTP A advanced HCC patients. Secondly, the risk of immune-related hepatotoxicity, which can potentially be life-threatening in patients with pre-existing liver dysfunction, suggests the acute need for more subtle clinical predictors [24].

To answer these questions, this multi-centre, international study was designed to explore the relationship between the ALBI grade and survival of patients with advanced HCC treated with ICI in Europe, North America and Asia.

## 2. Patients and Methods

### 2.1. Patients and Methods

Ethical approval to conduct this study was granted following review of the study protocol by the Imperial College Tissue Bank (R16008) and by local institutional review boards at each participating institution. The study was conducted in accordance with the principles stipulated in the Declaration of Helsinki and following Good Clinical Practice standards.

The study population derives from a multi-centre, prospectively maintained dataset of 341 HCC patients treated with ICI between 2016 and 2019. Patient data were identified from Oncology Pharmacy electronic records and entered into the dataset from 9 tertiary referral centres in the United States (*n* = 226), Europe (*n* = 68) and Taiwan (*n* = 47, Figure 1). All patients had a diagnosis of HCC based on imaging or histologically according to European Association for the Study of the Liver (EASL)/European Organisation for Research and Treatment of Cancer (EORTC) criteria [25].

Clinical data were obtained either at baseline, defined as the time of initial ICI treatment, or at cessation, defined as the time of discontinuation of the ICI treatment. Radiologic staging of the disease was performed using computerised tomography and/or magnetic resonance imaging as clinically indicated according to local practice. CTP functional class, performance status (ECOG PS) and the Barcelona Clinic Liver Cancer (BCLC) stage were computed for each patient according to standard pre-published methodology [26]. Response to ICI was evaluated according to RECIST criteria (version 1.1) by experienced radiologists, and best responses to ICI were used for the computation of the objective response (ORR) and disease control rates (DCR). As ORR to ICI were described to be similar when evaluated according to different criteria including mRECIST, we limited our analysis to RECIST 1.1.

We described treatment-emerging adverse events (AE) based on the Common Terminology Criteria for Adverse Events (CTCAE) classification, version 5.0.

The ALBI grade was calculated using the following equation: linear predictor = log_10_ bilirubin µmol/L × 0.66 + albumin × (−0.085). The linear predictor was further categorised into three different grades: grade 1 if ≤ −2.60, grade 2 if more than −2.60 and ≤ −1.39, and grade 3 if more than −1.39 as previously described [20]. The ALBI grade was calculated both at baseline and at ICI cessation using the corresponding bilirubin and albumin levels.

The primary clinical endpoint was overall survival (OS), calculated from the date of ICI initial treatment to the date of last follow up or patient’s death. As a secondary endpoint, we evaluated post-immunotherapy overall survival (PIOS), calculated from the date of permanent discontinuation of ICI to the date of last follow-up or patient’s death.

### 2.2. Statistical Analysis

Patient characteristics are presented as means or medians as appropriate. We performed univariable analysis of the different clinical factors potentially associated with patients’ survival using Kaplan-Meier methodology, followed by Log-rank test to evaluate the differences in median OS across prognostic strata. We further tested the independent prognostic value of each factor by multi-variable analysis using Cox regression models [21]. We utilised a stepwise backward procedure, with exclusion of variables with *p* value > 0.10. Pearson’s Chi-Square tests were performed to assess differences in proportions across groups. Receiver operating characteristic (ROC) curves were calculated for different prognostic factors and the area under the curve (AUC) method was used to compare the prognostic ability in predicting patients’ OS at landmark endpoints.

All statistical analyses were performed using SPSS version 25.0 (IBM Inc., Chicago, IL, USA) with all estimates being reported with 95% confidence intervals and a two-tailed level of significance of *p* ≤ 0.05.

## 3. Results

### 3.1. Baseline Patient Characteristics

Baseline characteristics of the studied population are presented in Table 1. The majority of patients were of BCLC C stage (*n* = 254, 75%) and CTP A class (*n* = 250, 73%). The most prevalent aetiologic factor for chronic liver disease was hepatitis C virus (HCV) infection (*n* = 135, 40%), followed by hepatitis B (HBV, *n* = 95, 28%). No data about the use of direct antiviral agents in patients with HCV infection were available. The majority of patients were treated with anti-PD(L)-1 monotherapy (*n* = 290, 85%). The ORR for the studied population was 20%, consisting of 23 complete responses and 42 partial responses, whereas the DCR was 54%. The median OS was 12.0 months (95% CI 9.2–15.0 months) and the median duration of follow-up was 11 months (range 1–34 months). The most common treatment-related AEs (trAEs), occurring at any grade, were liver toxicity (*n* = 59, 17%) and fatigue (*n* = 55, 16%). At the time of censoring, 252 patients had discontinued ICI treatment due to radiologically documented disease progression (*n* = 164, 65%), death (*n* = 27, 11%) or unacceptable toxicity (*n* = 14, 6%).

### 3.2. The Relationship between ALBI Grade and Clinico-Pathologic Factors

As shown in Table 2, when categorised according to ALBI grade, 104 patients classified as ALBI grade 1 (31%), 187 as grade 2 (55%) and 39 patients as grade 3 (11%). We subsequently evaluated the distribution of salient clinico-pathologic variables within each ALBI grade. Patients with advanced ALBI grade were more likely of male gender (*p* = 0.05) and cirrhotic (*p* < 0.001) secondary to viral hepatitis (*p* = 0.02). The presence of a more advanced ALBI grade prior to ICI commencement was significantly associated with a number of adverse clinico-pathologic features including more advanced CTP class (*p* < 0.001), BCLC stage (*p* < 0.001), ECOG PS (*p* < 0.001), higher alpha-fetoprotein (AFP) levels (*p* = 0.03).

ORRs stratified by ALBI were 25/98 for Grade 1 (25%), 29/177 for Grade 2 (16%) and 8/34 for Grade 3 (23%) and not dissimilar across groups (Pearson X^2^ 3.56, *p* = 0.16). Similarly, ORRs according to baseline CTP class were 50/237 (21%) for CTP A, 13/74 (17%) in CTP B and 2/7 (28%) in CTP C (Pearson X^2^ 0.72, *p* = 0.69) (Figure 2). In view of the increased likelihood of treatment-related toxicity with ICI combinations, we evaluated the distribution of trAEs by ALBI and CTP adjusting for the provision of single-agent PD-1/PD-L1 inhibitors versus combination regimens. The distribution of trAEs according to CTP and ALBI is described in Appendix A. As shown in Appendix A, we observed a trend towards statistical significance in the distribution of all-grade trAEs on the basis of CTP class and a significantly higher proportion of trAEs (56% vs. 42% vs. 29%, *p* = 0.01) across ALBI grades 1 to 3. However, following stratification by type of ICI regimen (mono vs. combination therapy), neither baseline CTP nor ALBI grade was associated with the occurrence of all-grade trAEs in our cohort (*p* > 0.05).

Lastly, we evaluated permanent discontinuation rates following unacceptable toxicity, which occurred in 9 CTP A (3%), 3 CTP B (1%) and 1 CTP C (0.3%) patients and in 4 ALBI 1 (1%), 7 ALBI 2 (2%) and 2 ALBI 3 (0.6%) patients, with no relationship with either CTP class (*p* = 0.51) nor ALBI grade (*p* = 0.92).

### 3.3. Pre-Treatment ALBI Grade Is an Independent Predictor of HCC Patients’ OS during ICI Therapy

In total, 330 patients had a quantifiable ALBI grade prior to ICI initiation and were eligible for OS analysis. When categorised according to the ALBI grade, median OS was 22.5 months (95% CI 18.5–26.4 months) in patients with ALBI grade 1 (*n* = 104, 31%), 9.6 months (95% CI 8.1–11.0 months) for grade 2 (*n* = 187, 57%), and 4.6 months (95% CI 2.3–6.8 months) for grade 3 (*n* = 39, 12%), respectively. Univariable analysis revealed significant differences in OS between ALBI grades (Log-rank *p* < 0.001, Figure 3A).

When stratified according to CTP class, median OS was 15.3 months (95% CI 11.3–19.3 months) in patients with CTP A, 7.5 months (95% CI 4.4–10.5 months) in CTP B, and 4.3 months (95% CI 2.4–6.1 months) in CTP C, with Kaplan-Meier analyses showing differences in OS between classes (Log-rank test *p* < 0.001, Figure 3B).

When considering solely patients with CTP A class and an available ALBI score (*n* = 239), we confirmed that the ALBI grade could further classify patients according to prognostically different strata as shown in Figure 3C. Patients in ALBI grade 1 (*n* = 100, 41%) had a median OS of 22.5 months (95% CI 18.5–26.4) compared to 10.8 months (95% CI 8.1–13.5 months) of grade 2 (*n* = 132, 55%) and 1.0 month (95% CI 0.9–2.8 months) for grade 3 (*n* = 7, 3%, Log-rank test *p* < 0.001). The distribution of ALBI grade across each CTP class is shown in Table 3.

We further investigated the prognostic validity of ALBI grade by multi-variable analysis adjusting for the potential confounding effect of other clinical factors. Alongside ALBI grade, univariable analyses of survival revealed CTP class (*p* < 0.01), BCLC stage (*p* = 0.04), AFP > 400 ng/mL (*p* = 0.02), achievement of radiologic disease control (*p* < 0.001) and provision of further anti-cancer therapy post-ICI (*p* < 0.001) as significant prognostic factors for OS. Multi-variable analysis of survival using Cox regression models confirmed worse ALBI grade as a significant predictor of inferior OS independent of other prognostic factors including achievement of disease control whilst on ICI (*p* < 0.001) and post-ICI therapy (*p* < 0.001) as shown in Table 4.

We evaluated the predictive ability of the ALBI grade versus CTP class and AFP in predicting 90-day mortality rate using ROC curve analysis and confirmed the superiority of the ALBI grade over CTP class with respective area under the curve values of 0.64 (95% CI 0.57–0.74, *p* = 0.001) for ALBI, 0.63 (95% CI 0.54–0.72, *p* = 0.006) for CTP and 0.63 for AFP (95% CI 0.54–0.71, Figure 3D). Lastly, we investigated whether dynamic changes of the ALBI grade from ICI commencement to discontinuation predicted for OS. We categorised patients displaying a 1- or 2-point improvement in ALBI (ALBI improved) grade against those showing a 1- or 2-point worsening (ALBI worsened) and those with stability across the two timepoints (ALBI stable), demonstrating progressively worse median OS across the 3 categories (median OS not reached and mean 25 months (95% CI 21.0–30.7 months) for ALBI improved versus 10.4 months (95% CI 8.8–11.9) for ALBI stable and 8.9 months (95% CI 8.5–12.2) for ALBI worsened, Log-rank *p* = 0.002, Appendix A).

### 3.4. The Relationship between ALBI Grade at ICI Cessation and PIOS

We further assessed the ability of the ALBI grade to predict for patients’ OS following permanent cessation of immunotherapy (PIOS). In total, 217 patients were eligible for PIOS analysis following exclusion of 85 patients who were actively receiving ICI treatment at the time of analysis and 28 patients with missing ALBI grade at immunotherapy cessation.

Within this group, 133 patients were evaluable for PS, the majority of whom had an ECOG PS score of 0–1 (*n* = 97, 45%). In total 78 patients (35%) received further anti-cancer treatment, mostly in form of tyrosine kinase inhibitors (*n* = 42, 20%). Median PIOS was 4.7 months (95% CI 3.6–5.7) and the total number of events was 131 (60%).

Following categorisation according to ALBI grade at cessation, median PIOS was 12.2 months (95% CI 8.9–15.5) for patients with ALBI grade 1 (*n* = 47, 26%), 5.2 months (95% CI 4.0–6.4) for grade 2 (*n* = 120, 66%) and 1.1 months (95% CI 0.8–1.4) for grade 3 (*n* = 50, 28%, Log-rank *p* < 0.001, Figure 3E). Within patients who, at ICI discontinuation, had preserved CTP A functional class and received further anti-cancer therapy (*n* = 73), those in ALBI grade 1 (*n* = 23, 31%) had a median OS of 12.2 months (95% CI 9.5–14.8) compared to 8.2 months (95% CI 5.3–11.2 months) of grade 2 (*n* = 40, 54%) and 5.0 months (95% CI 3.7–6.4 months) for grade 3 (*n* = 10, 15%, Log-rank test *p* < 0.001, Appendix A).

Other prognostic factors for PIOS included PS at immunotherapy cessation (*p* = 0.03) and provision of post-ICI therapy (*p* < 0.001, Table 5).

Multivariable analysis revealed a significant difference in PIOS between ALBI grades 3 and 1 (*p* < 0.001) but not between grades 2 and 1 (*p* = 0.18), a difference that was independent of post-ICI therapy status (*p* < 0.001).

## 4. Discussion

Unlike malignant melanoma, Merkel cell carcinoma and microsatellite-unstable cancers, where clinical responses to ICI are more than 40–50%, HCC is a moderately immune-sensitive disease, where ORRs are limited to 20% of patients [11].

Clinical trials of single-agent PD-1/PD-L1 inhibitors have failed to demonstrate the independent therapeutic value of ICIs in determining a significant OS improvement against placebo or standard of care [27]. Failure of these late-stage clinical development programmes is at least in part attributed to suboptimal disease stratification [28].

Whilst contemporary clinical trials of ICI have focused exclusively on CTP A patients to minimise the competing influence of cirrhosis on patients’ mortality, recent evidence suggests that the CTP classification may not fully capture more subtle heterogeneity in liver functional reserve. By simple combination of albumin and bilirubin, the ALBI grade has in fact shown prognostic superiority over CTP across the various BCLC stages and therapeutic modalities [21].

As the clinical development of ICIs continues unsupported by predictive correlates of response, use of the ALBI grade in ICI recipients might aid a more homogeneous patient stratification according to liver function and reduce its confounding effect over treatment-induced benefits on patients’ OS [29].

In our large, multi-institutional study, we demonstrated for the first time the independent ability of the ALBI grade in predicting the survival of HCC patients undergoing ICI treatment. In our patient cohort, the ALBI grade calculated prior to ICI commencement could significantly sub-classify patients with up to a 2-fold difference in median survival estimates across grades. Within the CTP A subgroup, which is at the focus of clinical trials of ICI in HCC, the ALBI grade could capture significant heterogeneity in outcome. Patients with an ALBI grade of 1 achieved a median OS in excess of 22 months, confirming the capacity of the ALBI to accurately identify a subset of HCC patients characterised by long-term stability of liver functional reserve and particularly favourable prognosis.

However, it should be kept in mind that attempts to validate a prognostic score might eventually prove unsuccessful when we consider external patients’ cohorts with clinical characteristics that diverge from the parental one. This was the case of the ART score (a score for the assessment of retreatment with transarterial chemoembolisation) whose prognostic performances could not be replicated within an independent cohort [30].

In our population, characterised for the most part by sorafenib-experienced patients, the extended survival observed in ALBI grade 1 patients compares favourably with the 26 months OS estimation registered for CTP A patients receiving sequential tyrosine kinase therapy [31].

Our landmark survival analysis using ROC curves shows that despite abandoning formal evaluation of coagulopathy, encephalopathy and ascites, measurement of liver function using the ALBI grade preserves non-inferior discriminative ability compared to CTP in estimating early mortality. This gives further credence to the view that albumin and bilirubin are the most influential factors of the CTP class and that computation of their prognostic value using the validated linear predictor first proposed by Johnson et al. may abate the so-called “floor” and “ceiling” effect from arbitrary categorisation of these variables.

Our findings confirm a significantly poorer survival in patients with deteriorating liver function such as CTP B patients. In fact, the risk-benefit ratio from treatments other than liver transplant in this challenging patient population is still under debate [32]. Whilst awaiting the results of prospective studies, we believe that similar considerations pertain to ICI as well, particularly in the setting of BCLC stage B patients, some of whom may be eligible for curative strategies [33].

Consistent with previously reported evidence, we found the ALBI grade to be significantly associated with a number of features that are predictive of adverse clinical outcome in HCC including more advanced BCLC stage, worse PS and higher AFP levels [34]. Interestingly, patients with better ALBI scores were more likely to have received immunotherapy combinations, a finding that is likely to reflect the tendency of treating physicians to prefer single-agent immunotherapy in patients presenting with worse disease status due to concerns over tolerance to synergistic immunotoxicity from combinations [35,36]. This imbalance is likely to explain the association between baseline ALBI and treatment-emerging toxicity in our whole study cohort, which, however, was not confirmed following adjustment for type of ICI regimen administered.

This is not the first study to highlight the prognostic importance of liver functional status in HCC patients treated with immunotherapy. A recent, single-centre study of 18 HCC patients treated with nivolumab mostly after sorafenib exposure (72%) and with a CTP score ranging between 7 and 9 has reported an ORR 17%, a median OS of 5.9 months and an incidence of trAEs of 28% [37].

We report an incidence of all-grade trAEs of 50% in CTP A and 34% in CTP B patients and respective median OS figures of 15.3 and 7.5 months, which, accounting for the presence of 14% of patients receiving combination immunotherapy in our series, is consistent and comparable with previous experience in this oncological indication. Whilst limited by small sample size, our study is the first multi-centre observational study to report on efficacy and safety data on ICI use in CTP C patients, where evidence is particularly limited [38]. Whilst CTP C patients were not dissimilar from CTP A/B on the basis of ORR (28%) and trAE rates (29%), the median OS of this patient group was significantly shorter in our study, 4.5 months, demonstrating the dominance of liver dysfunction over anti-tumour control in this patient subgroup. Whilst characterised by unavoidable heterogeneity in treatment and staging features including a small proportion of patients with BCLC D HCC who are suboptimal candidates for systemic treatment according to guidelines, our multi-institutional experience offers significant insight into the provision of immunotherapy in patients with more advanced degrees of liver dysfunction.

Advancing liver impairment is in fact intimately connected with progressive immune dysfunction within the hepatic immune microenvironment [39]. Whilst prospective clinical trials are evaluating the safety and efficacy of PD-1 inhibitors in patients with more advanced liver dysfunction (NCT01658878), our study provides initial retrospective evidence supporting homogeneity in radiologically demonstrated disease control across the whole spectrum of CTP or ALBI categories following ICI therapy.

In a previous study, we have demonstrated the ALBI grade to predict for post-sorafenib survival of 17.5, 7.5 and 1.9 months across ALBI grades 1–3 in patients eligible for second-line therapies [40]. Here, we have reproduced the potential of the ALBI to be utilised at the moment of permanent immunotherapy discontinuation, where an ALBI grade 1 predicted for a median OS time of over 12 months. In our study, we have also shown the ALBI grade to be a dynamic biomarker of liver functional reserve, with its deterioration from time to ICI commencement to discontinuation being harbinger of worse survival outcomes in patients with HCC.

With a rapidly increasing number of molecularly targeted and immune-based therapies available across different lines, optimal sequencing is essential to maximise survival outcomes in patients with advanced HCC [41]. Our data suggest potential for long-term survivorship in patients with a favourable ALBI grade even after tyrosine kinase inhibitor and immunotherapy failure, providing useful, evidence-based insight to enhance patient stratification and improve treatment sequencing in the clinic.

Our study acknowledges several limitations. The retrospective design and heterogeneity in treatment are inherent to the “real world”, observational nature of this study. However, all patients were recruited within experienced tertiary academic centres for the management of HCC, with treatment decisions being validated in the context of an individualised, multi-disciplinary discussion. The multicentre, geographically heterogeneous patient disposition of our sample limits statistical overfitting of our data and ensures external validity of our observations.

## 5. Conclusions

This study portrays safety and efficacy of ICI therapy across various degrees of liver dysfunction, including patient subgroups (i.e., CTP B/C) that are currently ineligible for ICI therapy in randomised clinical trials. Furthermore, we have validated the ALBI grade as a prognostic index in HCC patients treated with immunotherapy, highlighting its clinical utility both at initiation and cessation of ICI treatment. As the number of studies supporting its predictive ability across different stages and therapeutic modalities increases [21,22,40,42,43,44], consideration should be given to the ALBI grade as a prospective biomarker of hepatic functional reserve in routine clinical practice and in the clinical development of ICIs.

## Figures and Tables

**Figure 1 cancers-12-01862-f001:**
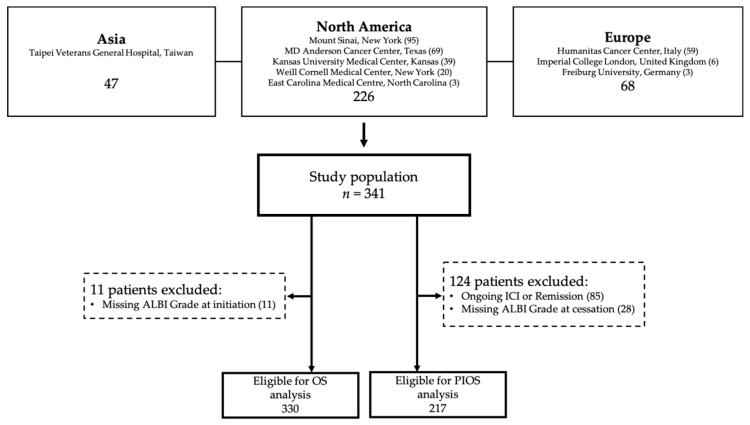
Study flow chart.

**Figure 2 cancers-12-01862-f002:**
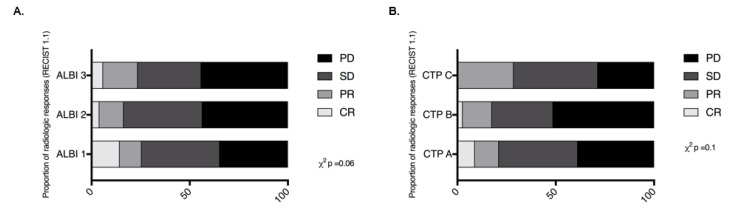
The relationship between ALBI grade (**A**) and Child-Turcotte-Pugh (CTP) class (**B**) and radiologic disease control by RECIST 1.1 criteria in patients with hepatocellular carcinoma (HCC) treated with immune checkpoint inhibitors (ICI).

**Figure 3 cancers-12-01862-f003:**
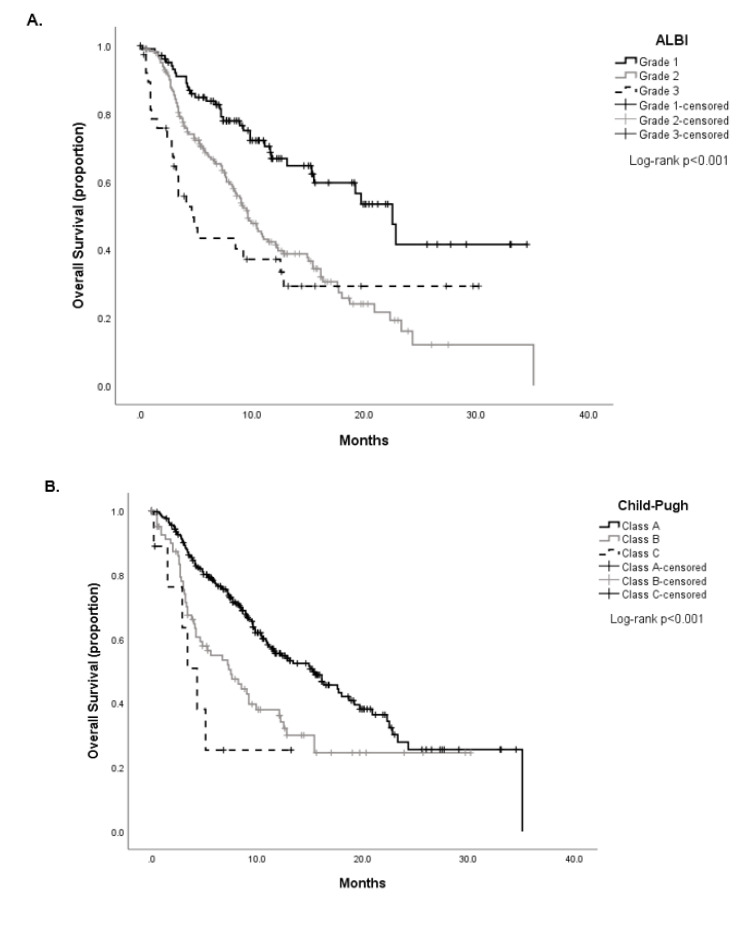
Prediction of overall survival by ALBI grade and CTP class in unselected HCC patients treated with ICI (*n* = 330, **A**,**B**) and in patients fulfilling CTP A criteria (*n* = 239, **C**). Receiver operating characteristic (ROC) curve analysis demonstrating the role of ALBI, CTP class, and AFP in predicting 90-days mortality from ICI commencement (**D**). The prognostic role on post-immunotherapy survival (PIOS) of the ALBI grade at the moment of ICI cessation (**E**).

**Table 1 cancers-12-01862-t001:** Clinical characteristics of the studied patient cohort.

Characteristic	*n* = 341 (%)
Age in Years	
Median (Range)	64 (15–89)
**Gender**	
Male	262 (77)
Female	79 (23)
**Cirrhosis**	
Present	242 (71)
Absent	99 (29)
**Risk factor for Liver Disease**	
Hepatitis B Infection	95 (28)
Hepatitis C Infection	135 (40)
Alcohol Excess	57 (17)
Non-Alcoholic Steato-Hepatitis (NASH)	34 (10)
Other	15 (4)
**Child-Turcotte-Pugh Class**	
A	250 (73)
B	81 (24)
C	9 (3)
**Barcelona Clinic Liver Cancer Stage**	
A	5 (2)
B	72 (21)
C	254 (75)
D	10 (3)
**AFP** (ng/mL)	
Median (Range)	115 (1–1,148,416)
**Albumin** (g/L)	
Median (Range)	36 (12–53)
**Bilirubin** (millimol/L)	
Median (Range)	14.5 (3–210)
**ALT** (IU/L)	
Median (Range)	45 (0–272)
**ALP** (IU/L)	
Median (Range)	135 (26–1064)
**Platelet count**	
Median (Range)	161 (42–670)
**Maximum Diameter of Largest Lesion** (**cm**)	
Median (Range)	5.2 (0.6–21.5)
**Extrahepatic Spread**	
Absent	166 (49)
Present	175 (51)
**Treatment for HCC**	
Resection	103 (30)
Ablation	62 (18)
Transarterial Chemoembolisation	156 (45)
Radio-Embolisation	81 (24)
External Beam Radiotherapy	34 (10)
Sorafenib	207 (61)
Other Systemic Therapies	32 (9)
**Prior Lines of Anti-Cancer Treatment**	
1	129 (38)
2	183 (54)
>2	29 (8)
**Immunotherapy Treatment**	
Anti-PD(L)-1 Monotherapy	290 (85)
Anti-PD(L)-1 + CTLA-4 Combination	25 (7)
Anti-PD(L)-1 + TKI Combination	24 (7)
Anti-CTLA-4 Monotherapy	2 (1)
**ALBI Grade**	
1	104 (31)
2	187 (55)
3	39 (11)
Missing	11 (3.2)
**ALBI Grade at Cessation**	
1	47 (14)
2	120 (35)
3	50 (15)
Missing	124 (36)

**Table 2 cancers-12-01862-t002:** The relationship between baseline albumin-bilirubin (ALBI) grade and salient clinico-pathologic variables.

Characteristic	ALBI G1 (*n* = 104) (%)	ALBI G2 (*n* = 187) (%)	ALBI G3 (*n* = 39) (%)	*p* Value
**Viral Aetiology**	57/47	122/64	30/8	0.02 *
Y/N	(55/45)	(66/34)	(79/21)	
**Gender**	71/33	148/39	33/6	0.05 *
M/F	(68/32)	(79/21)	(85/15)	
**Cirrhosis**	52/52	149/38	34/5	<0.001 *
Y/N	(50/50)	(73/27)	(89/11)	
**Child-Turcotte-Pugh class**	101/2/1	134/52/1	7/25/7	<0.001 *
A/B/C	(97/2/1)	(72/16/1)	(18/64/18)	
**BCLC Stage**	0/33/70/1	3/29/153/2	2/10/20/7	<0.001 *
A/B/C/D	(0/32/67/1)	(2/16/82/1)	(5/26/51/18)	
**AFP**	73/30	101/79	19/17	0.03 *
<400/>400 ng/mL	(71/29)	(56/44)	(53/47)	
**ECOG PS**	103/1	177/10	31/8	<0.001 *
0–1/2–3	(99/1)	(95/5)	(80/20)	
**ICPI treatment**	79/25	164/23	39/0	0.001 *
Monotherapy/Combination	(76/24)	(88/12)	(100/0)	
**Disease control rate**	64/40	100/87	19/20	0.28
CR + PR + SD/PD + NE	(62/39)	(54/47)	(49/51)	
**Toxicity as Reason for Discontinuation of ICI** Y/N				0.79
	4/99	7/179	2/37	
	(4/96)	(4/96)	(5/95)	
**Overall Survival (Months)**				<0.001 *
Median (95% CI)	22.5 (18.5–26.4)	9.6 (8.1–11.0)	4.6 (2.3–6.8)	
**Immunotherapy Duration (Months)**				<0.001 *
Median (95% CI)	5.1 (6.3–9.0)	3.3 (4.3–5.7)	2.3 (3.4–8.2)	

* indicates statistical significance at *p* < 0.05.

**Table 3 cancers-12-01862-t003:** The distribution of ALBI Grade in each CTP group.

CTP	ALBI Grade	*n* (%)	Median Survival (95% CI)
CTP A (*n* = 242)	**ALBI 1**	101 (42%)	22.5 (18.5–26.4)
	**ALBI 2**	134 (55%)	9.6 (8.1–11.0)
	**ALBI 3**	7 (3%)	4.6 (2.3–6.8)
	**Overall**		15.3 (11.3–19.3)
CTP B (*n* = 79)	**ALBI 1**	2 (3%) 52 (66%) 25 (32%)	-
	**ALBI 2**		7.2 (4.5–9.8)
	**ALBI 3**		8.5 (1.5–15.4)
	**Overall**		7.5 (4.4–10.5)
CTP C (*n* = 9)	**ALBI 1**	1 (11%) 1 (11%) 7 (78%)	-
	**ALBI 2**		-
	**ALBI 3**		3.4 (1.3–4.2)
	**Overall**		4.3 (2.4–6.1)

**Table 4 cancers-12-01862-t004:** Uni- and multi-variable analysis of survival in patients with HCC treated with ICI.

Variable	Patients (*n* = 330)	Univariable HR (95% CI)	*p* Value	Multi-Variable HR (95% CI)	*p* Value
**ALBI Grade**,	103/185/39				
2 vs. 1		2.2 (1.5–3.2)	<0.001	2.1 (1.4–3.0)	<0.001 *
3 vs. 1		2.8 (1.6–4.8)	<0.001	3.1 (1.8–5.4)	<0.001 *
**Age**, <65/>65	189/149		0.66		
**Gender**, M/F	259/79		0.80		
**Aetiology**, Viral/Non-Viral	215/121		0.34		
**Child-Turcotte-Pugh**,	247/81/9				
B vs. A		1.8 (1.3–2.5)	<0.001		
C vs. A		3.4 (1.5–7.8)	0.004		
**BCLC Stage**, C + D/A + B	260/78	1.5 (1.0–2.3)	0.04		
**Baseline AFP**, >400/<400 ng/mL	128/198	1.4 (1.1–2.0)	0.02		
**ICPI Regimen**, Monotherapy/Combination	287/51		0.61		
**Post-ICPI Therapy**, Active Treatment/BSC	102/205	0.55 (0.40–0.77)	<0.001	0.30 (0.20–0.48)	<0.001 *
**Disease Control Rate**, CR + PR + SD/PD + NE	185/153	3.32 (2.44–4.52)	<0.001	4.88 (3.43–6.96)	<0.001 *

* indicates statistical significance at *p* < 0.05.

**Table 5 cancers-12-01862-t005:** Uni- and multi-variable analysis of PIOS in patients with HCC treated with ICPI.

Characteristic	Patients (*n* = 217)	Univariable HR (95% CI)	*p* Value	Multivariable HR (95% CI)	*p* Value
**ALBI Grade at Cessation**,	47/120/50				
2 vs. 1		2.0 (1.1–3.4)	0.01 *	1.5 (1.0–3.5)	0.18
3 vs. 1		5.1 (2.8–9.3)	<0.001 *	3.9 (1.9–8.0)	<0.001 *
**Age,** <65/>65	115/102		0.53		
**Gender**, M/F	167/50		0.26		
**Aetiology**, Viral/Non-Viral	128/89		0.94		
**ECOG PS at Cessation** 2–4/0–1	51/120	1.7 (1.1–2.6)	0.03 *		
**Baseline AFP**, >400/<400 ng/mL	81/131		0.06		
**ICPI Regimen**, Monotherapy/Combination	180/37		0.39		
**Post-ICPI Therapy**, Active Treatment/BSC	78/139	0.3 (0.2–0.5)	<0.001 *	0.3 (0.2–0.5)	<0.001 *

* indicates statistical significance at *p* < 0.05.

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
