# Peer review of "Immunotherapy in Hepatocellular Cancer Patients with Mild to Severe Liver Dysfunction: Adjunctive Role of the ALBI Grade"

_cancers, 2020, doi:10.3390/cancers12071862_

Round 1

Reviewer 1 Report

The authors have adequately revised the manuscript.

I have no further comment.

Reviewer 2 Report

Dear Authors,
Your paper, in the final version, is very interesting and I hope it can be published in this Journal.

This manuscript is a resubmission of an earlier submission. The following is a list of the peer review reports and author responses from that submission.

Round 1

Reviewer 1 Report

In this manuscript, the authors examined ALBI score to predict the prognosis of HCC patients receiving immunotherapy. They showed that the ALBI score is superior to a conventional predictive score for liver reserve function, Child-Turcotte-Pugh (CTP) score. This study established the importance and usefulness of the ALBI score for the immunotherapy of HCC.

Major:
  1. PIOS, when stratified according to CTP class, should be analyzed.
  2. Does the change in ALBI scores between prior to and after ICI treatment predict OS? And is it useful to evaluate the therapeutic effects of ICIs?
  3. Previous study (PMID: 31940757) showed that AFP is also a good predictive factor for ICI treatment of HCC patients. The ROC analysis of AFP should be performed.

Minor:
  • lines 177-180. P values are not the same as Table 2. Are these analyzed by different methods?

Reviewer 2 Report

SPECIFIC COMMENTS

The affiliations are not correctly ordered. Personeni and Rimassa are indicated as 4,5 (4 Medical Oncology and Hematology Unit, Humanitas Cancer Center, Humanitas Clinical and Research 17 Center - IRCCS, Via Manzoni 56, 20089 Rozzano, Milan, Italy; 5 Department of Biomedical Sciences, Humanitas University, Via Rita Levi Montalcini 4, 20090 Pieve 19 Emanuele, Milan, Italy) and should therefore listed in a different order in the Authors list or assigned to a different number, following authors intent.

ABSTRACT

The background section is not clear enough to properly introduce the study. It is necessary to further improve this section. The remaining sections result appropriate and no further revisions are needed.

KEYWORDS

Some of the keywords (HCC; albumin-bilirubin grade; ALBI; immunotherapy) are not represented in the MeSH Browser. It would be advisable to choose keywords present on MeSH Browser in order to make the paper easier to be identified by stakeholders.

INTRODUCTION

The introduction is interesting to read and complete, although some clarifications could be made.

In Page 2 line 72, it should be considered that in the era of “tailored medicine” several evidences underlined how also Sorafenib therapies should to be “tailored” on patients in order to maximize the therapeutic effects and to reduce any possibility of adverse reactions, without forgetting that adverse events in patients with hepatocellular carcinoma treated with sorafenib have prognostic significance [J Hepatol. 2019 Dec;71(6):1175-1183 ---- Therap Adv Gastroenterol. 2016 Mar;9(2):240-9]. The authors could briefly analyze these points in relation to the previously reported papers, in this section or in the discussion, as they retain more appropriate.

At Page 3 line 112 the use of impersonal prepositions is preferred to personal because the introduction should remain neutral and should not be “personalized” from the beginning by the Authors.

PATIENTS & METHODS

At Page 3 line 126, more informations should be provided about criteria or clinical characteristics determining the use MRI rather than CT imaging or both.

The Authors stated that “Response to ICI was evaluated according to RECIST criteria (version 1.1).”. please, could the Authors explain why they did not used mRECIST?

Regardless of the adopted criteria for assessment of response to systemic therapies, patients should be evaluated by experienced radiologists to minimize variability in this critical instance [Eur Radiol. 2018 Sep;28(9):3611-3620.]. Could the Authors add (discuss) this point?

Could the Authors shift in the incipit of this section the paragraph: “Ethical approval to….”.

RESULTS

The most prevalent aetiologic factor for chronic liver disease was hepatitis C virus (HCV) infection (n=135, 40%). Did your patients undergo antiviral therapy with the new drugs (DAA)? It was demonstrated that the Delta of Liver stiffness is a useful non-invasive marker for predicting HCC development after DAA treatment [Dig Liver Dis. 2018 Jun;50(6):573-579.]. Have you recorded these data? Could the Authors discuss this theme in Discussion section?

Other results are clear and well-structured, and no modifications are necessary.

DISCUSSION

At page 11 line 276, Authors assumed that the study demonstrated the univocal ability of ALBI score in predicting the survival of HCC patients undergoing ICI treatments. However, other score systems did not work on all patient series and they may not prove completely objective tools, as demonstrated in previous experience on other scores [Dig Dis. 2014;32(6):711-716]. It will be advisable to discuss this point, highlighting the possible need to validate these results in a large monocentric series as an external cohort.

At page 12 line 304 the study underlines the importance of liver functional status in HCC patients treated with ICI. This topic could be further analysed in the discussion, focusing on this important issue in the recent dedicated literature. In fact, it is widely known that in patients with hepatocellular carcinoma and Child-Pugh-Turcotte class B (CPT-B), owing to borderline liver function, any intervention might be offset by liver function deterioration [Lancet Oncol. 2017 Feb;18(2):e101-e112.]. Please, could the Authors discuss the potential of this treatment related to the previous argument reported in the [Lancet Oncol. 2017 Feb;18(2):e101-e112.].

Furthermore, it is well known that the treatment type is an independent prognostic factor in BCLC-B patients and curative options (that not represent the standard of care for BCLC-B patients) offer the best outcome [Liver Int. 2017 Mar;37(3):423-433.]. Therefore, TACE (the standard of care for BCLC-B patients) probably could be can be replaced by other therapies especially in those patients with poor liver function. According to the result of the present study, the Authors believe that this ICI treatment could have a place in the future in this patient population (BCLC-B patients with labile liver function) in relation to the limits reported in the literature [Liver Int. 2017 Mar;37(3):423-433.]? In this case, the Authors can add some lines for discussion.

If the final discussion will be longer than now after the proposed suggestions, the Authors could reduce/remove the first two-three paragraphs (lines 261-272), which sound more for an introduction.

REFERENCES

Page 15 line 449 the reference 23 DOI is improperly underlined (?).

SUPPLEMENTARY MATERIALS

The same issue of authors list is present also in supplementary materials. Authors affiliations have to be re-ordered correctly.
